# Observations of Archaeological Proxies through Phenological Analysis over the Megafort of Csanádpalota-Juhász T. tanya in Hungary Using Sentinel-2 Images

**Athos Agapiou** [1,*] **, Alexandru Hegyi** [2,3] **and Andrei Stavilă** [4]

1   Earth Observation Cultural Heritage Research Lab, Department of Civil Engineering and Geomatics, Faculty of Engineering and Technology, Cyprus University of Technology, Limassol 3036, Cyprus
2   Centre for Southeast Asian Studies, Kyoto University, 46 Shimo-Adachi, Yoshida, Sakyo-ku, Kyoto 606-8501, Japan
3   Applied Geomorphology and Interdisciplinary Research Centre (CGACI), Department of Geography, West University of Timișoara, 300223 Timișoara, Romania
4   Faculty of Letters, History and Theology, West University of Timișoara, Vasile Pârvan Blvd. No.4, 300223 Timişoara, Romania
*   Correspondence: athos.agapiou@cut.ac.cy

**Abstract:** This study aims to investigate potential archaeological proxies at a large Bronze Age fortification in Hungary, namely the Csanádpalota–Juhász T. tanya site, using open-access satellite data. Available Sentinel-2 images acquired between April 2017 and September 2022 were used. More than 700 images (727) were initially processed and filtered, accounting at the end of more than 400 (412) available calibrated Level 2A Sentinel images over the case study area. Sentinel-2 images were processed through image analysis. Based on pan-sharpened data, the visibility of crop marks was improved and enhanced by implementing orthogonal equations. Several crop marks, some still unknown, were revealed in this study. In addition, multi-temporal phenological observations were recorded on three archaeological proxies (crop marks) within the case study area, while an additional area was selected for calibration purposes (agricultural field). Phenological observations were performed for at least four complete phenological cycles throughout the study period. Statistical comparisons between the selected archaeological proxies were applied using a range of vegetation indices. The overall results indicated that phenological observations could be used as archaeological proxies for detecting the formation of crop marks.

**Keywords:** archaeological proxies; phenological observations; archaeological prospection; Sentinel-2; crop marks; vegetation indices; Hungary

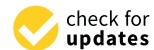


## 1. Introduction

Archaeological proxies have been widely used in the past for detecting areas with potential interest [1]. The discovery of subsurface archaeological remains through archaeological proxies is usually applied by detecting the so-called "crop marks" [2,3]. Crop marks occur when healthy vegetation, such as barley or cereal crops, is cultivated over shallow buried archaeological remains. Therefore, the crops are either stressed (negative crop marks phenomenon) or enhanced (positive crop marks phenomenon) [2,4], and this can be used—once detected by remote sensing sensors—as a first indication (proxy) of the existence of buried remains.

The identification of crop marks is usually carried out by analyzing high-resolution satellite and airborne imageries. At the same time, ground spectroradiometers have also been tested successfully [5,6]. While the use of sub-meter satellite sensors is widely adopted [7,8], limitations still exist regarding (a) the use of medium-resolution datasets

(e.g.,10 m, 20 m pixel resolution) and (b) the analysis of multi-temporal phenological observations of crop marks [4,9,10]. In addition, the use of earth observation sensors in several areas around the globe is still fragmented [11].

The investigation of remote sensing techniques for studying prehistoric sites is becoming increasingly appealing to scholars since it can provide valuable information in less time than systematic archaeological excavations, which usually take a long time to be completed. Of course, remote sensing in archaeology cannot replace field archaeology, but it can assist archaeologists in better understanding the sites even before excavations. While advanced remote sensing archaeology has a long record in some parts of the world (see [11,12]), in Eastern Europe this started to be popular only recently. The launch of the Google Earth platform, released in the beginning of the 21st century (2001), was a true revolution in viewing archaeological sites, as it was throughout the world [13–15]. Before this, the perspective from above for archaeological sites was entirely determined by existing aerial images. The use of aerial photographs in eastern European archaeology is now considered a standard procedure [16–35].

Megaforts are the most recognizable prehistoric sites in Eastern European archaeology. Megaforts, as Antony Harding refers to them [36], are Bronze Age fortifications built at the confluence of two major rivers, Mureș and Tisa. Because of their size, these sites quickly became the subject of archaeological investigations, both non-invasive and invasive. In the study [37], one of the first remote sensing papers describing the impressive fortification from Cornești-Iarcuri in Romania, which is considered to be Europe's largest Bronze Age fortification, was published. In 2008, a follow-up paper was published [38]. Because of its impressive size, the Cornești-Iarcuri fortification quickly gained international attention, and other remote sensing techniques were used [39–43]. Another mega fort, Sântana, was discovered in southwestern Romania, north of the Mureș River. Remote sensing, including light detection and ranging (LiDAR) and ground geophysics, as well as comprehensive archaeological excavations, were also used to investigate the Sântana–Cetatea Veche site [44,45]. A more recent paper was published on the use of more advanced satellite image remote sensing techniques to study the sites of Cornești–Iarcuri and Sântana–Cetatea Veche [46].

The mega-fort of Gradište Iđoš in northern Serbia was also studied using remote sensing tools in 2020 [47]. In Hungary, two large fortifications stand up: Orosháza-Nagytatársánc and Csanádpalota–Juhász T. tanya. The latest is discussed in this paper. Only a little research has been done on Orosháza-Nagytatársánc [48,49], while Csanádpalota–Juhász T. tanya was investigated more thoroughly [50–53]. Many other large bronze-age fortifications were discovered within the region by photo interpretation of Google Earth images or other satellite images followed by verifications in the field [54,55].

The European space program, namely the Copernicus program, providing free and open access satellite optical datasets, like those of Sentinel 2A and 2B sensors, opened a new era in the scientific domain of remote sensing archaeology [4,9,56,57]. Images can be downloaded, through big data cloud hubs, in a very short period after the acquisition time, in a calibrated and corrected form, minimizing radiometric, atmospheric, and geometric distortions. Nevertheless, skepticism still exists in their use due to the medium resolution of the image provided (10- and 20-m resolution).

The scope of this study is twofold: from one hand, to utilize novel approaches for detecting archaeological proxies in an area limitedly studied in the past, in Csanádpalota-Juhász T. tanya in Hungary, increasing the archaeological visibility and understanding of the area, blended with existing knowledge. On the other hand, the study aims to present the importance of phenological observations—a topic limited discussed in the literature—for detecting archaeological proxies [58,59]. Satellite observations and phenological observations through the analysis of hundreds of Sentinels 2 images over the area of the Csanádpalota-Juhász T. tanya site are reported here.

## 2. Case Study

The Bronze Age fortification of Csanádpalota-Juhász T. tanya [60] located in the Great Hungarian Plain, south of Csanádpalota (Csongrád county), near the Romanian–Hungarian border, was selected as a case study (Figure 1). The mega fort of Csanádpalota-Juhász T. tanya, along with other sites of its kind like Cornești–Iarcuri, Sântana–Cetatea Veche, and Gradište Iđoš, is one of the most significant Bronze Age sites, not only because of its size but also because it is the expression of social paradigm change and represents the transition to another socio-cultural environment, where the power is shown by these massive structures. Between 2011 and 2013, the site was studied through rescue excavations carried out during the motorway construction connecting Szeged to the Hungarian–Romanian border [50–53,60]. Ditches and trash pits are archaeological features from the Bronze Age. Some ditches were "V" or "U" shaped with a depth of 2–3 m and a width of 4–7 m; others were smaller (the trace of the ditches can be seen in Figure 2). Ceramic vessels and artefacts made of bronze or horns were deposited in pits lying on the bottom of the north-south orientated ditch. The ditch was approximately 6–7 m wide and 1.5–2 m deep. After the ditch was abandoned, deliberate deposits were made [50–53,60].

These fortifications are dated to the Late Bronze Age (between the end of the 16th century and the 10th and 9th centuries BCE) based on archaeological findings. At the same time, excavations revealed the construction of defense systems such as earth ramparts, palisades, and ditches, and the existence of violent conflicts that resulted in their destruction [45,61,62]. It is up to future research to ascertain how these fortifications were internally organized and to reconstruct the material culture of the communities that inhabited them.

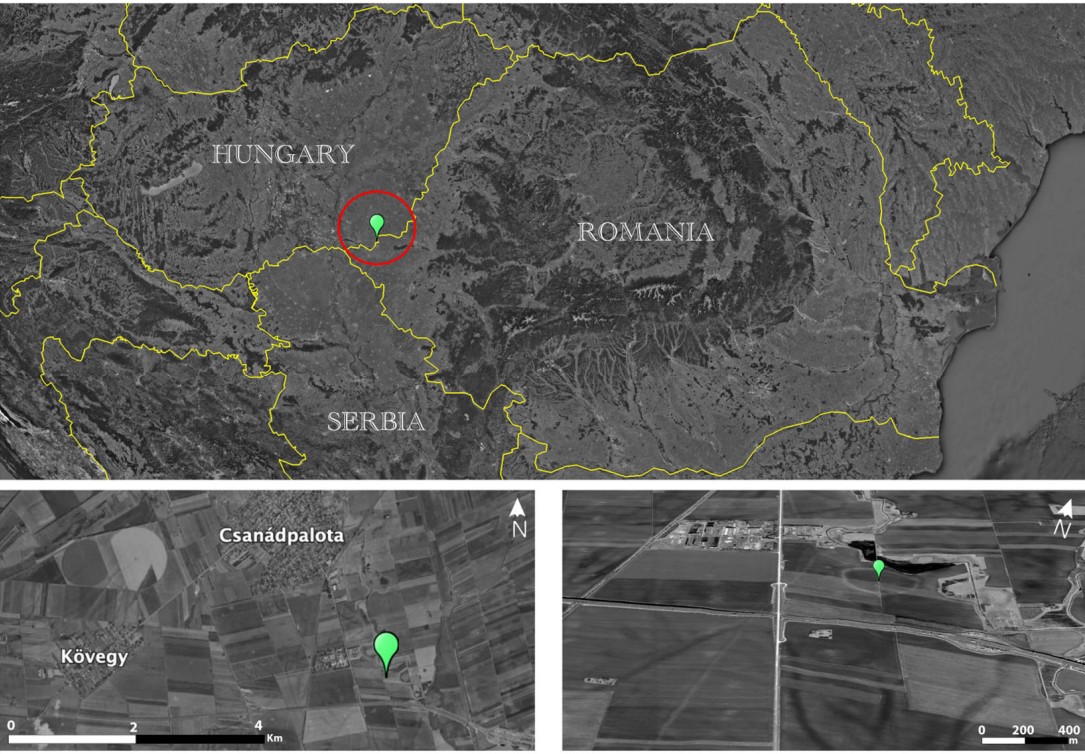

**Figure 1.** Location of Csanádpalota-Juhász T. tanya (Google Earth Image—edited).

The circular enclosure uncovered in the fortification's core region, known as Földvár, was also explored during the rescue excavations. In this case, archaeological excavations were preceded by systematic archaeological material collecting, core drilling, aerial photography, and magnetic prospections. The rampart structure, which consisted of a strip of 30–40 cm of burnt adobe, was identified in the trench excavated here. Postholes were discovered behind it, most possibly forming a palisade. Two 3 m deep "V" shaped ditches

located in front of the rampart were investigated. These ditches have also yielded significant archaeological material [50–53].

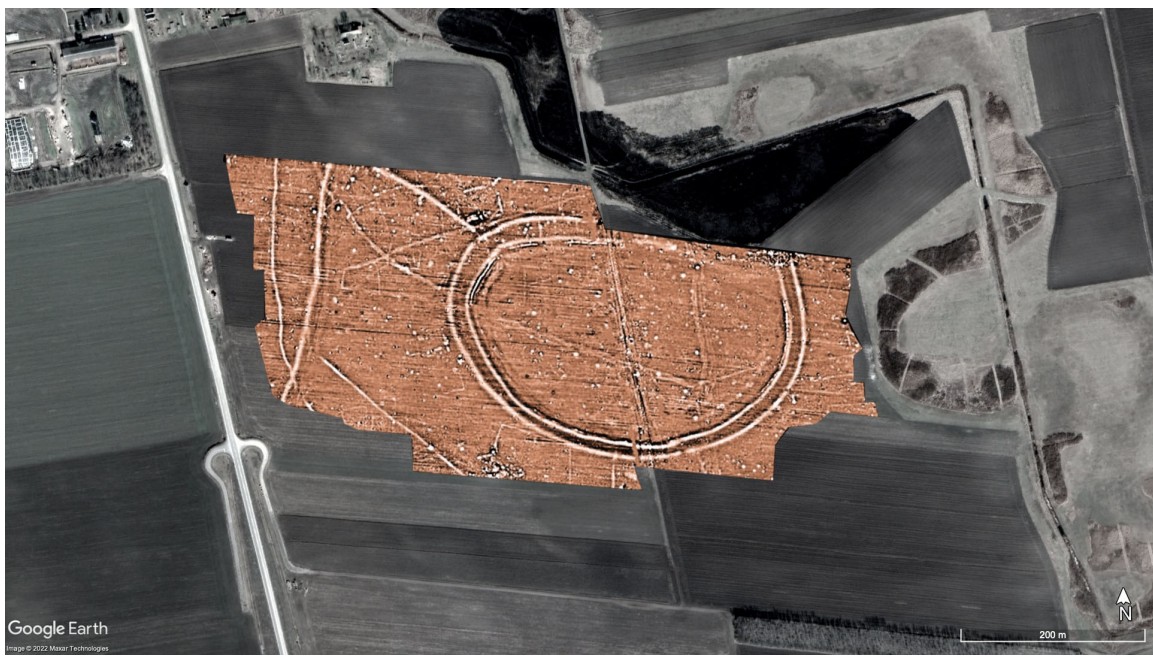

**Figure 2.** The magnetic map of Csanádpalota-Földvár enclosure (after [53] Figure 3). This figure was created by georeferencing the image published in [53] Figure 3. The extracted RGB image was enhanced on the red band to better highlight the archaeological features. Visualisation dynamics −/+ 10 nT.

## 3. Methodology

### 3.1. Satellite Image Processing

In our study, 727 Sentinel 2A and 2B images available during the period from April 2017 until September 2022 were initially queried in the QGIS 3.26 environment using the "Semi-automatic classification" plugin [63]. Afterwards, by applying cloud coverage-image quality thresholds, the total Sentinel 2 dataset was dropped by half (57%), enabling us to proceed with 412 Sentinel 2 images. Figure 3 shows the total number of available Sentinel images per month, after the implementation of the cloud coverage and quality thresholds. The maximum number of Sentinel 2 images per month was 13 (August 2018), while the minimum was 0 (April 2017). For all months, except for April 2017, at least a single image was available over the case study area. The average number of images per month was estimated to be around 6.15 and a median value of 6.

The Sentinel images were retrieved from the Copernicus Open Access Hub (previously known as Sentinels Scientific Data Hub) as a Level-2A product (Bottom Of Atmosphere, BOA, reflectance images). Each Level-2A product is composed of $100 \times 100$ km$^2$ tiles in cartographic geometry (UTM/WGS84 projection). Sentinel 2A and 2B optical sensors can record electromagnetic radiation in 13 spectral bands, with a range from the visible range to the shortwave infrared (SWIR) at a spatial resolution (ground sampling) of 10 and 20 m.

Then, individual Sentinel 2 images were downloaded and further analyzed in the ArcGIS Pro environment to enhance archaeological proxies. Different enhancement approaches were applied, including radiometric histogram enhancements, vegetation indices (see Table 1), pan-sharpening techniques (Gram–Schmidt method), and orthogonal transformations adopted by [46,64–66]. The overall results were compared with existing knowledge of the area.

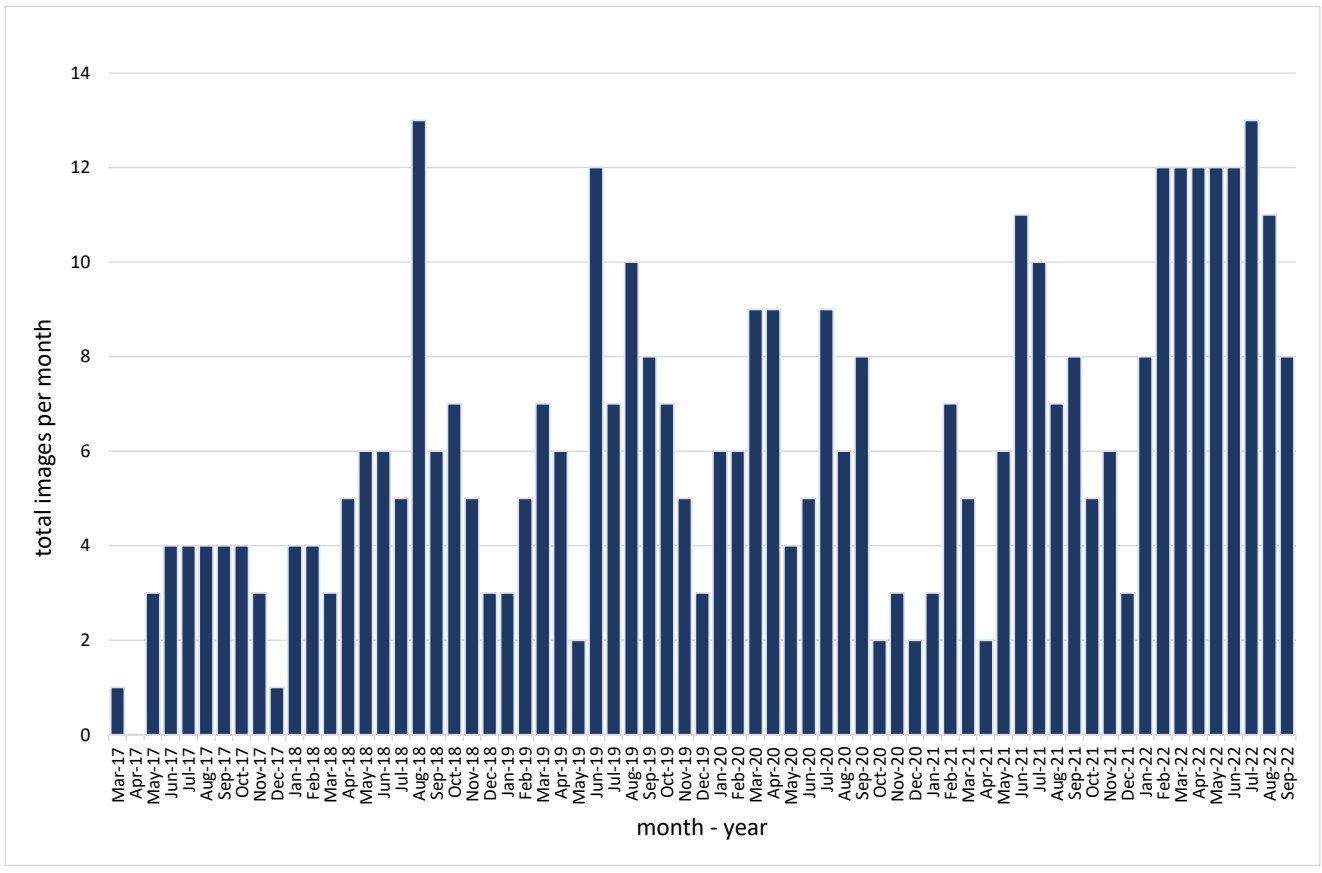

**Figure 3.** Total Sentinel 2 (Level 2A) used in this study per month (period from April 2017 until September 2022).

**Table 1.** Linear correlation coefficient of the four areas using the ARVI profiles.

| Vegetation Index Name | Veg. Index | Equation | Reference | Equation |
|---|---|---|---|---|
| Atmospheric Resistance Vegetation Index | ARVI | $(pNIR - prb)/(pNIR + prb)$, $prb = pred - \gamma\,(pblue - pred)$ | [67] | (1) |
| Enhanced Vegetation Index | EVI | $2.5\,(pNIR - pred)/$ $(pNIR + 6\,pred - 7.5\,pblue + 1)$ | [68] | (2) |
| Normalized Difference Vegetation Index | NDVI | $(pNIR - pred)/(pNIR + pred)$ | [69] | (3) |
| Soil Adjusted Vegetation Index | SAVI | $(1 + L) \times (pNIR - pred)/$ $(pNIR + pred + L)$ | [70] | (4) |
| Soil and Atmospherically Resistance Vegetation Index | SARVI | $(1 + 0.5)\,(pNIR - prb)/$ $(pNIR + prb + 0.5)$ $prb = pred - \gamma\,(pblue - pred)$ | [71] | (5) |
| Tasseled Cap—Greenness | TCG | $-0.28482 \times B02 - 0.24353 \times B03 -$ $0.54364 \times B04 + 0.72438 \times B08 +$ $0.084011 \times B11 - 0.180012 \times B12,$ | [72] | (6) |

### 3.2. Analysis of Phenological Profiles

More details regarding phenological observations were investigated for specific areas around the Csanádpalota-Juhász T. tanya site. Four different areas in the wider area of the site were analyzed based on their phenological profiles in the last five years. These areas are in relevant proximity, minimizing potential noise due to different types of soil, crop cultivation, and cultivation practice. Area 1 was selected in the outer circle of the

archaeological proxy of the site; area 2 was defined at the center of the circular enclosure; area 3 was over a linear feature observed in the northwest part of the circular enclosure; area 4 was at an agricultural field, as shown in Figure 4.

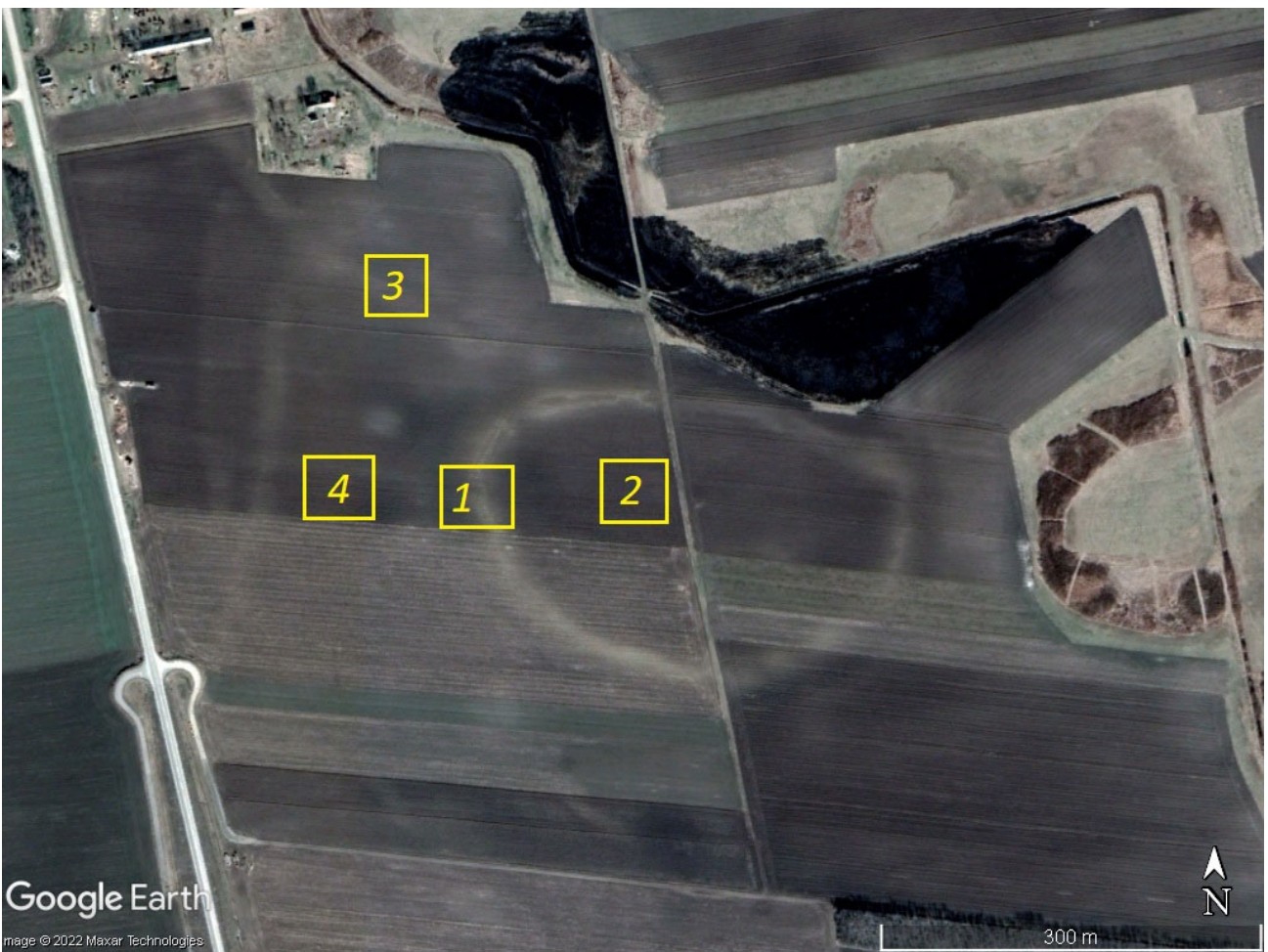

**Figure 4.** Selected areas of interest at the Csanádpalota-Juhász T. tanya site. Area 1 is located in the outline of the circular enclosure of the site, area 2, at the center of the site, area 3 over a linear feature, and area 4 at a cultivated field close to the site (background high-resolution Maxar image of 4th March 2017, Google Earth).

In specific, utilizing the QGIS 3.26 environment and the "Semi-automatic classification" plugin [63], vegetation profiles were extracted based on the analysis of different vegetation indices. The latest included the Atmospheric Resistance Vegetation Index (ARVI), the Enhanced Vegetation Index (EVI), the Normalized Difference Vegetation Index (NDVI), and the Soil Adjusted Vegetation Index (SAVI). In addition, the Tasseled Cap transformation was retrieved for the three components of brightness, greenness, and wetness. The equations for these indices and the Tasseled Cap are provided in Table 1.

By employing time-series vegetation indices over a specific area of interest, phenological observations can be made, as shown in Figure 5. The NDVI index was explored to average over a period and map "normal" growing phenological conditions of the area of interest for a given time of year. Higher NDVI values were observed during the development and mid-season periods, while lower values were observed at the initial or late stages of the vegetation phenology. The latest can also be reported in cases of stressed conditions. Indeed, phenological observations can be further elaborated to describe vegetation's health or even understand any stress conditions and vegetation changes.

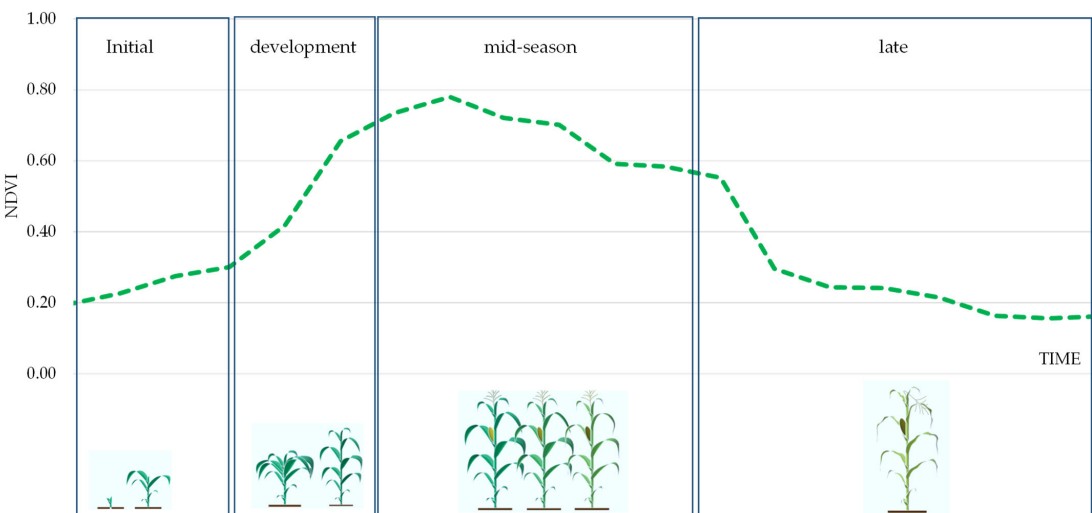

**Figure 5.** Phenological diagram of crops at different stages of development along with the NDVI profile.

## 4. Results

### 4.1. Sentinel-2 Image Processing

Individual Sentinel-2 image satellite processing was carried out. Due to the complexity of the local geomorphological context of the site, which is in an area with several paleochannels, interpreting archaeological findings using medium-resolution satellite images can be challenging. The implementation of vegetation indices (see Table 1) in Sentinel-2 images did not yield satisfactory results, as shown in Figure 6. This was also reported in other references in the past [73–75].

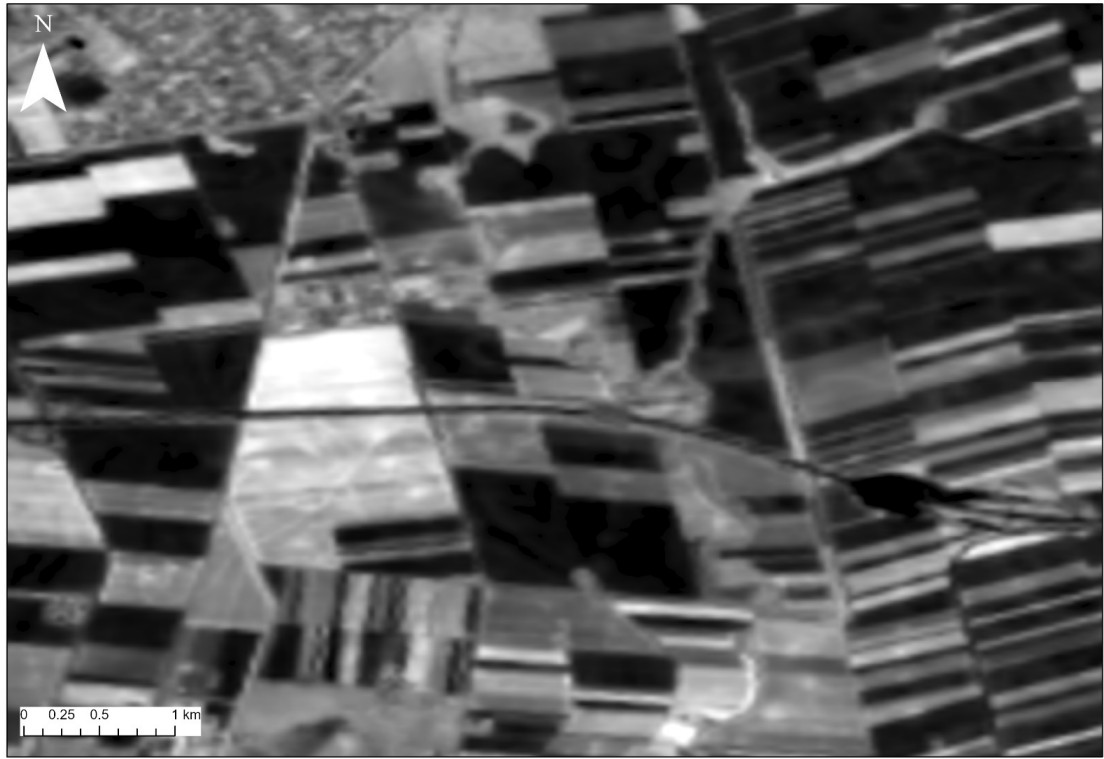

**Figure 6.** NDVI index applied at Sentinel-2 image at the Csanádpalota-Juhász T. tanya site. No obvious archaeological proxies can be spotted in the area.

Therefore, it was decided to improve the spatial resolution of the 20 m spectral bands of the Sentinel-2 image to 10 m resolution using the Gram–Schmidt pan-sharpening method [76] (Figure 7, top). In addition, orthogonal transformations [64–66] were applied to Sentinel-2 images to enhance the presence of crop marks (Figure 7, bottom) and further support image interpretation. The spatially improved 10-m resolution Sentinel image allowed us to locate other fortification features north of the circular enclosure which can be cross-referenced with better quality images to provide a better image of defensive elements distribution (see Figure 8, left).

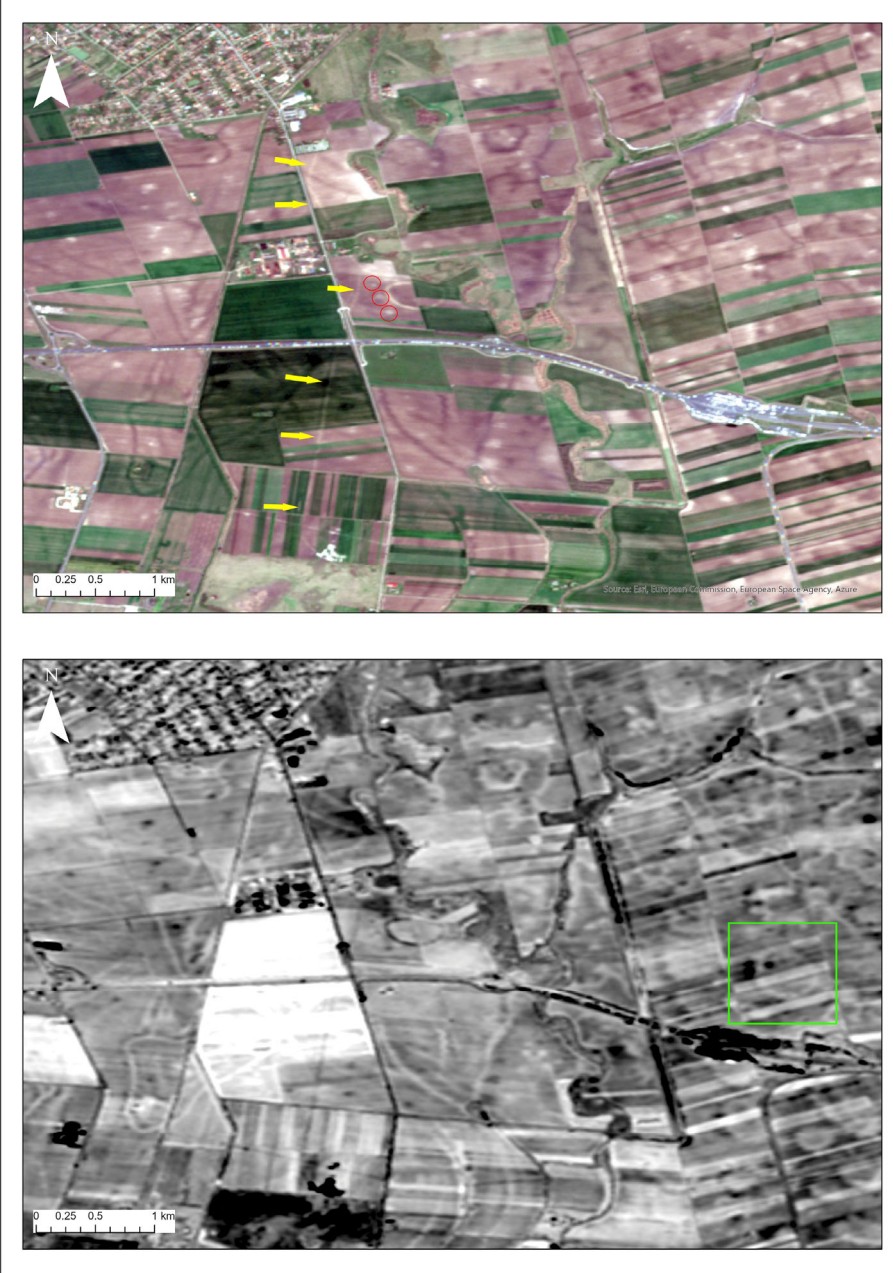

**Figure 7.** Gram–Schmidt pan-sharpening Sentinel-2 image over Csanádpalota-Juhász T. tanya (**top**); results after the implementation of the orthogonal transformation (**bottom**). Specific areas shown with arrows, circles, and squares are discussed in the paper.

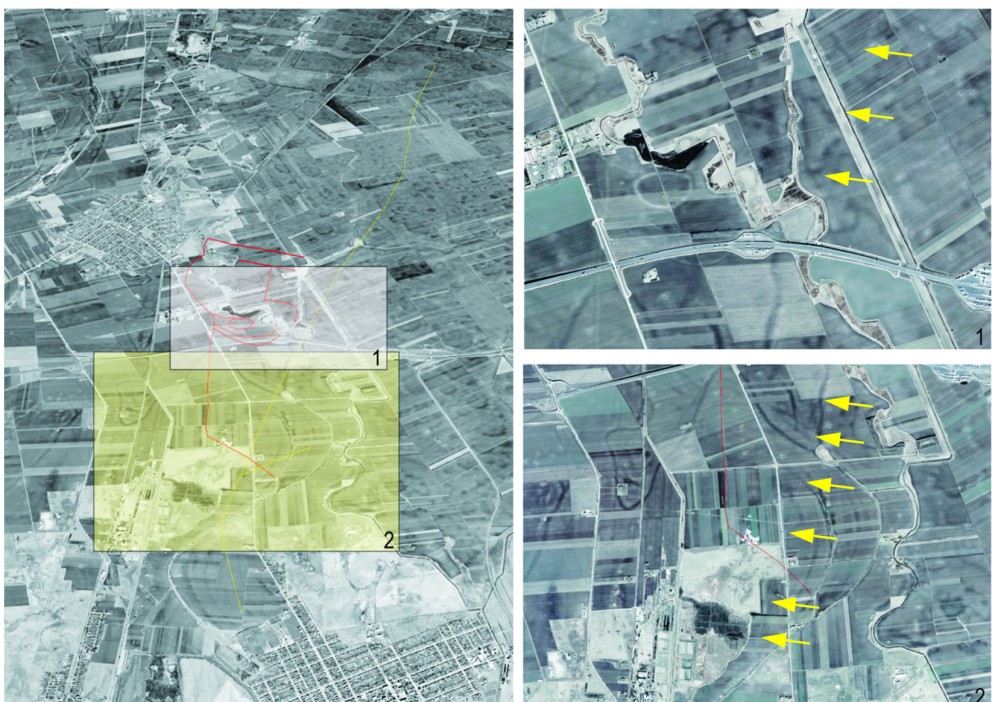

**Figure 8.** Edited March 2017 satellite image (available in Google Earth Pro). Left: the red lines denote easily traceable features; the yellow line denotes a linear feature identified in archaeological papers on the fortification's south-eastern part. The yellow line runs approximately 13 km between Nădlac and Mezhegyes. Right: Rectangles 1 and 2 show details of the regions highlighted on the left image—the yellow arrows indicate the linear feature.

As in the case of other similar investigations at sites in the wider area of Cornești and Sântana (see [46]), the analysis of satellite images related to Csanádpalota-Juhász T. tanya sparked several discussions and revealed novel archaeological features that will be confirmed by archaeologists in the future, given that only the filed survey can confirm or refute the current image of the fortification.

Pan-sharpening and orthogonal transformation (Figure 7) for Sentinel 2 satellite images, for example, highlight the presence of at least three interesting structures west of the main circular enclosure (Figure 7, red circles). When the position of these structures is compared to the existing magnetic map, the multispectral anomalies are caused by a number of rectangular archaeological features located within that area. The closest analogy for this multispectral signature is at Sântana–Cetatea Veche, where the central structure, which the magnetic map confirmed to be a large building, had a similar multispectral signature on Sentinel 2 images [46]. On pan-sharpened images, the western linear fortification system is well individualized (Figure 7, yellow arrows), whereas the presumed eastern system is completely invisible to multispectral analyses. However, local geomorphology played an important role on the eastern side—a north–south paleochannel runs through that area and could have served as protection for the eastern side. We believe that this idea could be explored further in the future by using a more elaborate geoarchaeological approach to determine the age of the paleochannel. The orthogonal transformation of Sentinel 2 revealed another circular structure 2 kilometers east of the site's main enclosure, which is most likely a natural feature (Figure 7, green square). These circular depressions could occur in this fluvial landscape, but they should be confirmed in the field as well.

Nevertheless, the fortification system is still visible on Sentinel images—the central part, the western and northern ramparts. The fortification line, which is depicted in archaeological papers to the east ([51], Figure 4; [53], Figure 9), is not visible at all. The lack of any crop marks related to that fortification line was peculiar and intriguing, and it

necessitated further investigation on more detailed images to explain why this feature was completely absent.

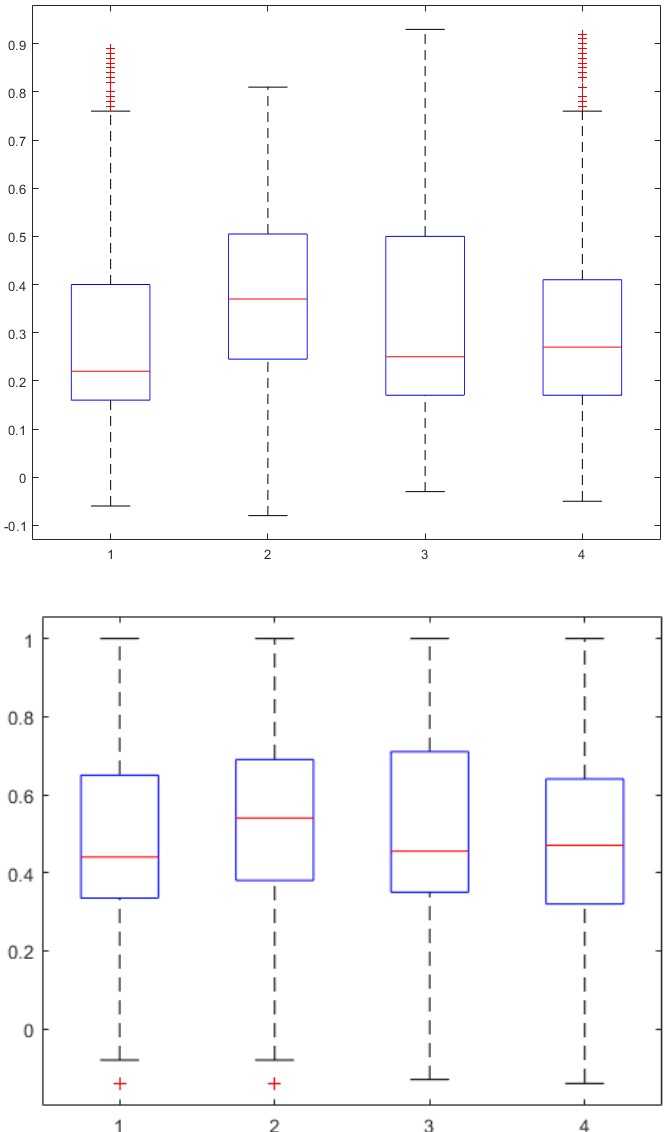

**Figure 9.** (**Top**) Boxplot of the NDVI values of the four areas of interest over the Csanádpalota-Juhász T. tanya site. The *X*-axis corresponds to the selected area (area 1 is located in the outline of the circular enclosure of the site, area 2, at the center of the site, area 3 over a linear feature, and area 4 at a cultivated field in close proximity to the site), while *Y*-axis corresponds to the NDVI value (range −1 to +1). (**Bottom**) Boxplot of the same areas based on the ARVI index.

In Figure 8, a satellite image from March 2017 (available in Google Earth Pro) shows that the so-called eastern fortification line is a distinct feature that is not demanded to be in contact with the main fortification. Given that many of these linear features of considerable length can also be traced in the Romanian Banat, it may be a linear defensive earthwork. What is more intriguing is that this linear feature has a length of around 13 kilometers and can be traced from Nădlac to Mezőhegyes (Figure 8, yellow line), which leads us to believe it could be a medieval road which could have speculated the higher ground of a former rampart or element related to the Bronze Age fortification. Only future archaeological surveys will be able to determine the location of the eastern fortification line, if one exists.

*4.2. Phenological Observations*

Further to image analysis, phenological observations for the four areas (see Figure 4) were applied. The results are shown in the boxplots in Figure 9. The boxplot on top of Figure 9 indicates the median NDVI value (red line), the minimum and maximum score, and the first (lower) quartile, median, and third (upper) quartile. It is interesting to note that areas 1, 3, and 4 show comparable mean NDVI values, with different variances, while a difference is recorded for area 2. While the mean NDVI value is around 0.20–0.25 for areas 1, 2, and 4, the mean NDVI value for area 2 is around 0.40. Comparable results were retrieved for the four areas based on the ARVI index (Figure 9, bottom).

Areas 1 and 3 show similar patterns in contrast with areas 1 and 2 or areas 2 and 4. Indeed, if we use a trendline (linear regression model) to estimate the correlations between the different areas and the different indices used in this study (see Equations (1)–(6)), we can see a close similarity between areas 1 and 3. The linear correlation coefficient ($r^2$) for all different combinations of areas and vegetation indices is shown in Tables 2–6. EVI index (Equation (2)) provided a very low correlation and is not listed here.

**Table 2.** Linear correlation coefficient of the four areas using the ARVI profiles.

|  | Area 1 | Area 2 | Area 3 | Area 4 |
|---|---|---|---|---|
| **Area 1** |  | 0.72 | **0.89** | 0.79 |
| **Area 2** |  |  | 0.69 | 0.62 |
| **Area 3** |  |  |  | 0.78 |
| **Area 4** |  |  |  |  |

**Table 3.** Linear correlation coefficient of the four areas using the NDVI profiles.

|  | Area 1 | Area 2 | Area 3 | Area 4 |
|---|---|---|---|---|
| **Area 1** |  | 0.44 | **0.87** | 0.52 |
| **Area 2** | 0.44 |  | 0.46 | 0.23 |
| **Area 3** | 0.87 | 0.46 |  | 0.54 |
| **Area 4** | 0.52 | 0.23 | 0.54 |  |

**Table 4.** Linear correlation coefficient of the four areas using the SAVI profiles.

|  | Area 1 | Area 2 | Area 3 | Area 4 |
|---|---|---|---|---|
| **Area 1** |  | 0.44 | **0.88** | 0.52 |
| **Area 2** |  |  | 0.46 | 0.23 |
| **Area 3** |  |  |  | 0.54 |
| **Area 4** |  |  |  |  |

**Table 5.** Linear correlation coefficient of the four areas using the SARVI profiles.

|  | Area 1 | Area 2 | Area 3 | Area 4 |
|---|---|---|---|---|
| **Area 1** |  | 0.72 | **0.89** | 0.79 |
| **Area 2** |  |  | 0.69 | 0.62 |
| **Area 3** |  |  |  | 0.77 |
| **Area 4** |  |  |  |  |

**Table 6.** Linear correlation coefficient of the four areas using the TCG profiles.

|  | Area 1 | Area 2 | Area 3 | Area 4 |
|---|---|---|---|---|
| **Area 1** |  | 0.53 | **0.90** | 0.70 |
| **Area 2** |  |  | 0.47 | 0.34 |
| **Area 3** |  |  |  | 0.70 |
| **Area 4** |  |  |  |  |

If we consider the observations between profiles for areas 1 and 3, the $r^2$ is exceptionally high (>0.87) for all indices, indicating a strong correlation (and behavior) between the two areas. In contrast, the linear correlation coefficient between areas 1 and 2 is lower (0.44), while the correlation between areas 1 and 4 is estimated to be around 0.52. In addition, the correlation coefficient ($r^2$) for areas 2 and 3, as well as for areas 2 and 4, and 3 with 4, is calculated to be close to or lower than 0.50 for all indices.

Figure 10 shows the NDVI and ARVI plot matrixes for each one of the four areas (areas 1–4) using the whole dataset (412 images). The scatterplots are aligned with remarks made before (see Tables 2–6), as we can see that an almost linear correlation does exist between the two archaeological proxies over areas 1 and 3, in comparison, for instance, with results from areas 2 and 3 where the variance is more considerable. The diagonal rows of the matrices indicate the histogram of each area per vegetation index.

Phenological variations of all four areas selected during the study period (April 2017 until September 2022) were also studied. To better understand potential phenological variations due to the presence of archaeological proxies, one should recall the study of [5]. Non-homogenous areas with different soil compositions and environmental conditions can have a different impact on the phenological profile of a crop. Likewise, differences might occur due to different land management practices. In addition, as McCloy [77] argues, all these considerations lead to the hypothesis that the phenological curve—as detected from satellite sensors—can change in one of five ways (or a combination of them), as illustrated in Figure A1. In that study, the NDVI index was used to monitor phenological variations throughout time.

Figure 11 shows the phenological profile for the complete sample of the dataset (412 images). The *x*-axis displays the period based on the date of acquisition of the available images, while the *Y*-axis indicates the NDVI value. In addition, a moving average filter (2 periods) is applied (shown as lines in Figure 11) to smooth the sample. In particular, specific periods of interest can be considered those where the NDVI value is more significant than 0.2 (crops start to grow). Therefore, a value lower than 0.20 can be ignored, as these usually belong to other classes (e.g., soil, urban areas, and water bodies).

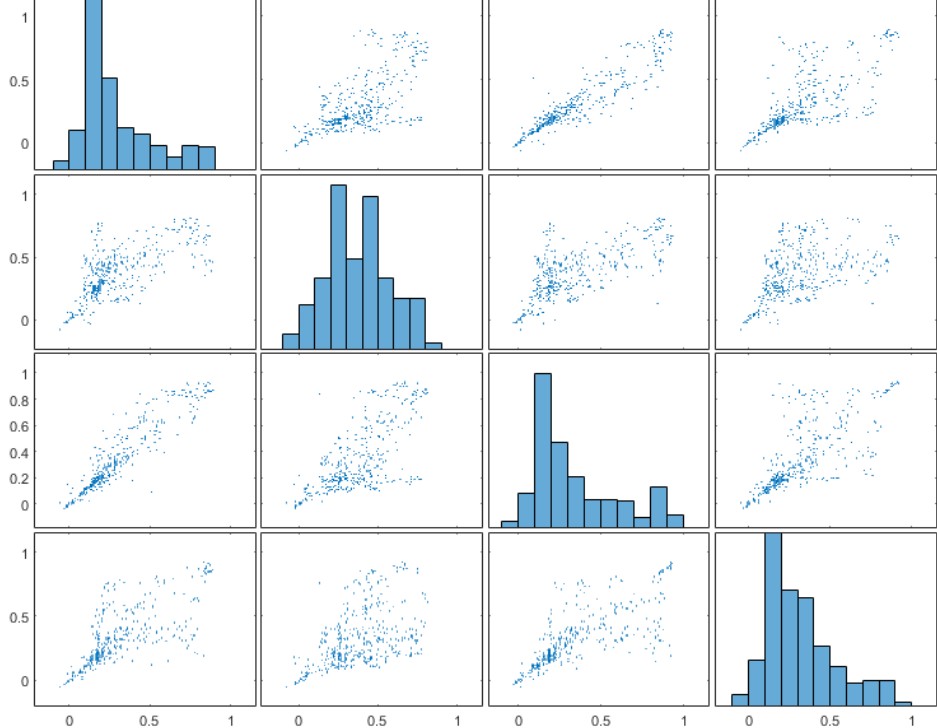

**Figure 10.** *Cont.*

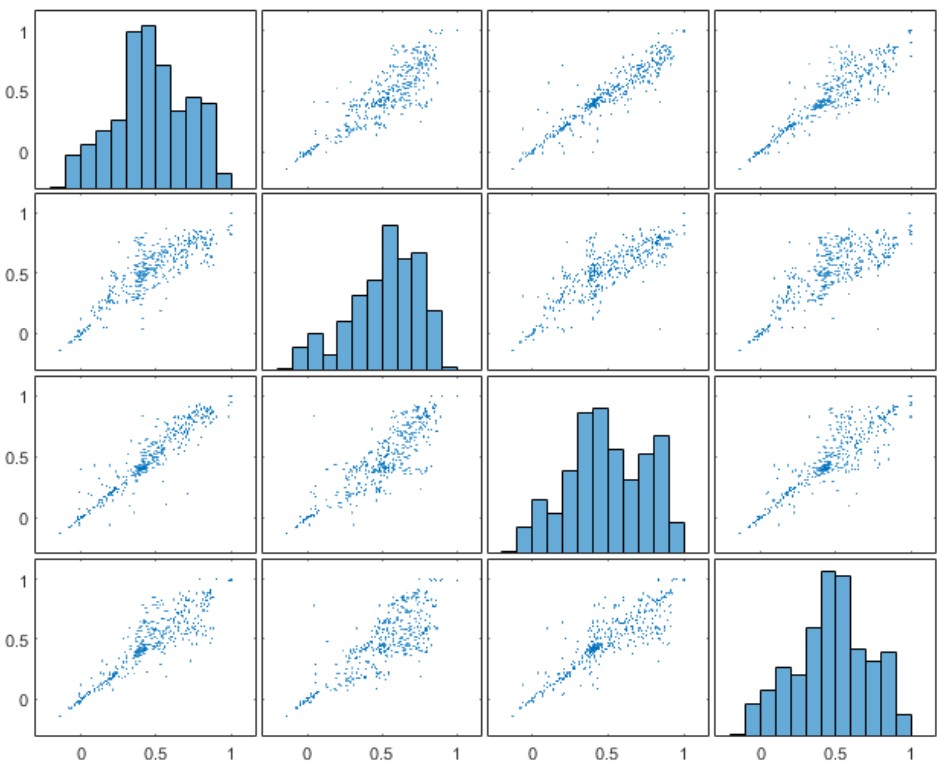

**Figure 10.** (**Top**) Plot matrix of the NDVI values of the four areas of interest over the Csanádpalota-Juhász T. tanya site. The *X*-axis corresponds to the selected area, while *Y*-axis corresponds to each one of the four areas. (**Bottom**) Plot matrix as before using the ARVI index.

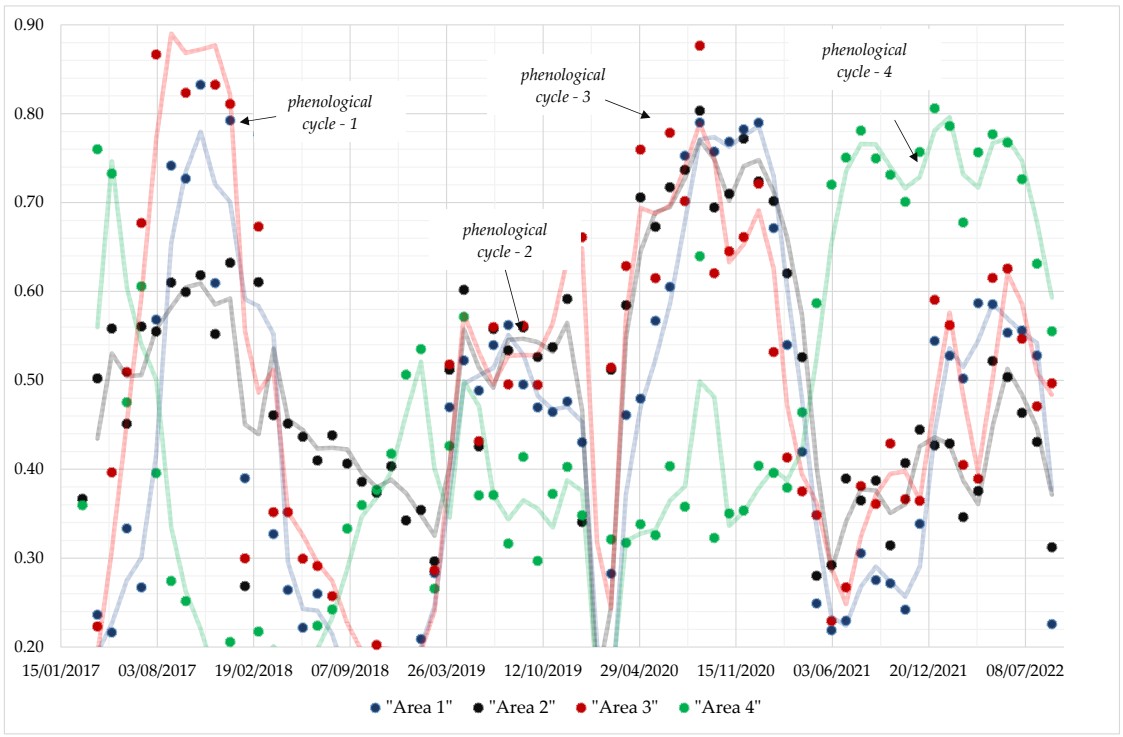

**Figure 11.** Phenological profiles are shown as dots over areas 1, 2, 3 (archaeological proxies), and 4 (cultivated area). A moving average filter of 2 periods (line) was applied to smooth the datasets.

To analyze the phenological profiles—as shown in Figure 11—two parallel studies can be made: (a) a first study may be focused on examining how crops' phenology is captured by the Sentinel sensors over archaeological proxies nearby, while (b) a second study can be focused on variations between areas with archaeological interest and "non-archaeological areas" (e.g., cultivated fields). For the study (a), a comparison between areas 1 and 3 was made. For this study (b), a comparison between areas 1, and 3 against area 4 was performed (see Figure 4).

As shown in Figure 11, differences between the NDVI values for all areas exist. The phenological cycle of the crops is well depicted at least four times for all areas. These cycles' profile is similar to the expected pattern and illustrated in Figure A1. In general, areas 1 and 3 (archaeological proxies, indicated with blue and red lines in Figure 11) tend to provide similar patterns, such as synchronous periods for the start and end of the phenological cycles and high peaks during the same periods. In addition, the area 2 profile is comparable with areas 1 and 3 for at least two phenological periods (March 2019 until April 2020 and April 2020 until June 2021). A distinct contrast to these measurements is those recorded over area 4 (non-archaeological area). While for the first two phenological periods, no comparison can be made against the other three areas, in the latest two phenological cycles, we can observe significant variations: during the third phenological cycle, we have for area 4 a high peak value at the same period as the other three areas (areas 1–3), indicating that the same cultivation practices were followed, and crops were planted. Nevertheless, the maximum NDVI value was up to 0.50 against to around 0.80 for the rest three areas under examination. A reverse phenomenon was captured during the last (4th) phenological cycle, whereas higher NDVI values were depicted for area 4 (around 0.80) and lower, but still high, values for areas 1–3.

## 5. Conclusions

This study presented the results of remote sensing analysis over the mega fort Bronze Age site of the Csanádpalota-Juhász T. tanya in Hungary. For this purpose, 412 Sentinel-2 images covering the period 2017 to 2022 were used. The analysis was carried out in two steps: firstly by analysing single Sentinel-2 datasets using various image processing techniques, and secondly by examining the phenological profile of specific areas around the site.

For the first step, although the 20 m resolution was too coarse to spot any crop mark formation, the pan-sharpening technique enabled us to improve the spatial resolution at 10 m and proceed with image enhancement techniques. The overall results of this analysis highlighted the vast bulk of the previously known archaeological features (part of the fortification lines). Nevertheless, our analyses revealed several previously unknown features in the archaeological literature but also sparked debate over the eastern fortification line's existence, which requires more investigations.

For the second step, interesting outcomes were generated concerning the potential use of phenological studies to detect crop marks. This topic still has limited discussion in the literature. Phenological similarities over archaeological proxies were reported (see areas 1 and 3), while the systematic spectral variation with non-archaeological areas (area 4) is also evident. This finding supports the hypothesis that studies dealing with the phenological profile of archaeological areas can be used as an indicator (proxy) for the existence of buried archaeological remains. Such studies must consider the proper use of multi-temporal analysis of remotely sensed data. Therefore, calibrated images provided by the Copernicus Programme seem ideal for the moment.

Future work can be considered, taking into consideration higher resolution datasets, including airborne and low altitude, to increase the archaeological visibility of the site further. At the same time, medium resolution interperion and analysis can be carried out to investigate other similar sites located on the broader landscape.

**Author Contributions:** Conceptualization, methodology, A.A. and A.H.; writing—original draft preparation, A.A. and A.H.; writing—review and editing, A.A., A.H. and A.S. All authors have read and agreed to the published version of the manuscript.

**Funding:** Part of the image processing and analysis in this study was funded by the ENSURE project (innovative survey techniques for detection of surface and sub-surface archaeological remains), a Cyprus University of Technology internal funding.

**Data Availability Statement:** Not applicable.

**Acknowledgments:** A.H. would like to acknowledge PN-III-P1-1.1-PD-2019-0939, founded by UE-FISCDI. A.A. would like to acknowledge the ENSURE project (Innovative survey techniques for detection of surface and sub-surface archaeological remains, CUT internal funding). The authors would like to acknowledge the use of open access and freely distributed satellite datasets from Copernicus Open Access Hub (for Sentinel 2 images).

**Conflicts of Interest:** The authors declare no conflict of interest. The funders had no role in the design of the study; in the collection, analyses, or interpretation of data; in the writing of the manuscript; or in the decision to publish the results.

## Appendix A

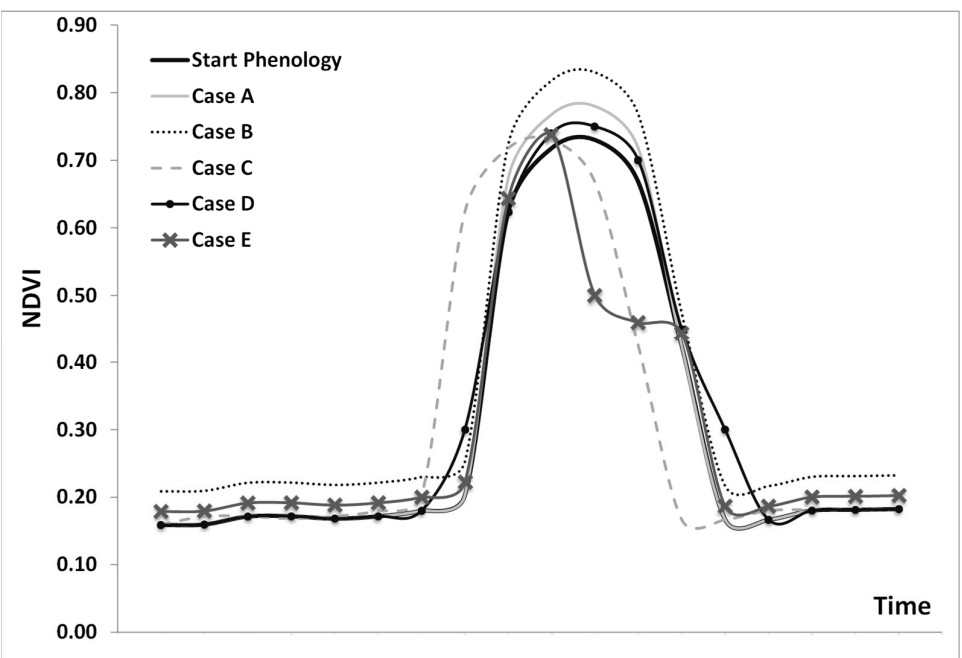

**Figure A1.** Phenological profiles under different conditions [5,77]. Case A: Change in the amplitude of the NDVI Phenological Curve in an area which is moisture constrained; a trend of increasing rainfall may result in an increase in peak greenness relative to the greenness out of the growing season (or the opposite which leads to a decrease of the peak greenness). Case B: Overall change of the greenness (NDVI) throughout the year for example in an area with similar (not the same) crop cultivation. Case C: Shift in the time of the peak in an area where changes in the temperature and/or rainfall regimes lead to a shift in the phenophase associated with growth, maturation and death. Case D: Lengthening of the growing season yielding a broader or narrower phenological trajectory through time as can occur with rising or falling temperatures in areas that are temperature constrained and lastly, Case E: Change in the shape of the phenological curve, during the phenological cycle in areas after the invasion of species or due to changes in the rainfall regime. On the other hand, this is also the case when the crop is characterized under stress due to the lack of soil nutrients. The latest might be related to buried archaeological remains such as walls (negative crop marks) ([5]).

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
