# Peer review of "Observations of Archaeological Proxies through Phenological Analysis over the Megafort of Csanádpalota-Juhász T. tanya in Hungary Using Sentinel-2 Images"

_remotesensing, doi:10.3390/rs15020464_

Round 1

Reviewer 1 Report

Journal Remote Sensing (ISSN 2072-4292)

Manuscript ID remotesensing-2090478

Title Observations of Archaeological Proxies through Phenological Analysis over the Megafort of Csanádpalota – Juhász T. tanya in Hungary using Sentinel-2 images

This reviewer finds the paper to be aimed at solving an important problem with an interesting method and suitable for publication in Remote Sensing. However, this reviewer has a few minor concerns regarding that the authors should address before the paper is accepted for publication. This reviewer recommends accepting the paper for publication after the authors address this reviewer’s minor concerns listed below.

Below please see my comments:

1.    Line 100. The explanation of the importance of the area in Bronze Age is inadequate. Add more information.

2.    The resolution of Figure 9 is poor

3.    ~ Add scale in figure 1

4.    Figure 11: change the red arrow color by another clear  

Reviewer 2 Report

The paper presents the application of remote sensing observations and the phenological analysis applied to a case study located in Hungary, namely the Megafort of Csanádpalota – Juhász T. tanya. Satellite images acquired by Sentinel 2 were retrieved, processed and used to extract phenological profiles of four test areas over the 2017-2022 period. The results show that phenological profiles can be used to individuate buried archaeological features.

Phenological analysis is out of my main expertise so I do not make any specific comment about it. The methods used to process satellite images are commonly used and apparently there are no advances for what concerns this topic. However, I have some doubts about the ordering of contents. Paragraph 3 (Methodology) is very short while paragraph 4 (Results) and its subparagraphs presents methodological aspects mixed with results. Methodology should clearly describe the methods and tools used for the two main topics (1.analysis of phenological profiles and 2.satellite image processing); this structure should then be replied for Results. The extraction of phenological profiles is described in few lines, while it could be better explained; a plot showing the results of the extraction from a single satellite image could be useful as well.

Moreover, I would consider to flip the order of the contents. Phenological observations are described before the image processing at present, but this appears to be misleading. The four test areas were localized on satellite images, and it's reasonable to use enhanced/processed images to better localize them before extracting the phenological profiles rather than later. This appears also in lines 150-155 of paragraph 3, where the different enhancement approaches, later described in lines 296-308, are stated before the extraction of phenological profiles.

I would also suggest to review the English language.

You can find more specific comments below:

line 19-20: please give her the exact number of images;

line 34: it is arguable that archaeologists have “recently” explored archaeological proxies for detection of site;

lines 55-56: distinguish between Google Earth, Google Earth Pro and especially Google Earth Engine, as they are different services;

line 57: the reference to the ‘advent of the Digital Globe’ is not clear, please explain better;

lines 61-65: the bibliographic style should be formatted accordingly to the journal rules in every part of the text;

lines 76-77: ‘from some remote sensing work’, please rephrase with a less generic meaning;

line 91: the first paragraph could be into subparagraphs that help the reader (e.g. State of the Art, Scope);

line 96: consider also Agapiou A, Lysandrou V, Lasaponara R, Masini N, Hadjimitsis DG. Study of the Variations of Archaeological Marks at Neolithic Site of Lucera, Italy Using High-Resolution Multispectral Datasets. Remote Sensing. 2016; 8(9):723. https://doi.org/10.3390/rs8090723 and Conesa FC, Orengo HA, Lobo A, Petrie CA. An Algorithm to Detect Endangered Cultural Heritage by Agricultural Expansion in Drylands at a Global Scale. Remote Sensing. 2023; 15(1):53. https://doi.org/10.3390/rs15010053 for the use of archaeo proxies and phenological observations;

line 110-111: is the pottery deposition important for the history of the site and for the paper? If yes, explain why, otherwise it can be omitted and fig. 2d removed;

line 112: maybe an archaeological drawing of the entire area, as those already published in the references 51 (fig. 9) or 53 (fig. 4), would help;

line 134: please avoid phrases like ‘more than …’ here and everywhere else in the paper, just state the exact number; if you want to specify that images come from Sentinel 2A and 2B satellites please specify that the platforms have the same sensor; give also a brief overview of the sensor characteristics;

line 135: QGIS is mentioned here, but it is not clear how it is used for the paper purposes; how did you query the images? Why did you use QGIS? Did you use any plugin? Please specify;

line 137: simply give the number of used images;

lines 142-143: are both the average and median values significant? Otherwise just give the most representative;

line 160: please avoid ‘etc.’ here and everywhere else in the paper;

169-171: were the vegetation profiles extracted for all the 412 images? Pleas provide more details;

line 188: please join the two images in one;

lines 191-193: there is no need to repeat ‘(area 1 is located … proximity to the site) here and also in the caption of fig. 6;

line 219: see the comment for lines 188 and 191-193. In addition, the scatterplots here are not clearly readable: please find a way to improve their visibility;

line 226: there is no need to repeat ‘(areas 1 to 4)’;

lines 229-252: you refer to ‘one or more of five ways’ the phenological curve can change, but then you mention only five cases; however, cases A to E are similar to McCloy 2010 p. 2446 and identical to Agapiou – Hadjimitsts 2011 053554-10 and 11. You could try to better rephrase the cases adapting them to the case study. Otherwise a bibliographic reference would be enough. In any case, please better specify the bibliographic source;

line 253: the chart is identical to the one published from reference 5; is it necessary here? If yes, where do the data come from? Is it possible to plot a similar chart for the case studies discussed in the paper?

line 257: there is no need to specify ‘per case study (area 1 to area 4)’;

lines 268-269: there is no need to repeat the localization of areas;

lines 268-269: ‘Area 1 is located…linear feature’ is not necessary, the reference to fig. 4 is enough;

line 288: the chart is a bit confusing; lines would probably be better; the legend on top is confusing as well because it looks like if it refers to the four spikes of the phenological cycle; moreover the legend is only for points and not for lines;

line 302: this is a bit ambiguous, because some Sentinel 2 bands have a spatial resolution of 10 m; please rephrase; adding the sensor characteristics as suggested for line 134 will help;

line 317: ‘existing magnetic map’, is it the one shown in fig. 2a? If yes, please always use the same term (‘magnetic map’ or ‘magnetometry’) and change everywhere accordingly;

line 332: please increase the size of red arrows in the image; as you named subfigures with letters in fig. 2 consider to do the same for image 10 and 11;

line 336-341: it is not clear at what features you are referring to; the reference to fig. 2 is unclear, which letter and what archaeological papers? Please add references if needed; a general plan of the site would help (comment for line 112);

lines 360-365: this part appears to be unnecessary and can be removed or rephrased; at line 362 give the exact number of images;

lines 386-388 and 393-397: in both parts the author A. S. is missing; please add the author’s contribution;

The structure of the paper should be changed in a clearer way before publishing.

Here you can find some typos:

line 47: please order the references ‘[4, 9-10]’;

line 48: ‘(see [11])’ > ‘[11]’?

line 64: ‘become’ > ‘became’?

line 86: ‘opens’ > ‘opened’?

line 185: missing space between ‘1’ and ‘2’, ‘2’ > ‘3’?

line 200: ‘([eq. 2))’ > ‘([eq. 2])’;

line 234 ‘[from 5]’ > ‘[5]’;

line 261: ‘these are usually other classes’ > ‘these usually belong to other classes’?;

line 265: ‘(b) while a’ > ‘while (b) a’?

line 269: missing space between ‘1’ and ‘3’;

line 276: is ‘picks’ for ‘peaks’?

line 282: is ‘pick’ for ‘peak’?

line 328: ‘reveals’ > ‘revealed’?

References:

ref. 8 'imagerty' > 'imagery'

ref. 20-21 missing spaces

ref. 27 'coments' > 'comments'

Reviewer 3 Report

What I did not find extremely clear in the introduction was your switch of focus from the entire site to more specific areas within the site. This could proabably be made more clear to those who are not familiar with this region/site. Any way, congratulations on this interesting paper (esp. phenological profiles).
